# PI3K Inhibitors in Advanced Breast Cancer: The Past, The Present, New Challenges and Future Perspectives

**DOI:** 10.3390/cancers14092161

**Published:** 2022-04-26

**Authors:** Paola Fuso, Margherita Muratore, Tatiana D’Angelo, Ida Paris, Luisa Carbognin, Giordana Tiberi, Francesco Pavese, Simona Duranti, Armando Orlandi, Giampaolo Tortora, Giovanni Scambia, Alessandra Fabi

**Affiliations:** 1Department of Woman and Child Health and Public Health, Division of Gynecologic Oncology, Fondazione Policlinico Universitario Agostino Gemelli IRCCS, 00168 Rome, Italy; margherita.muratore@guest.policlinicogemelli.it (M.M.); ida.paris@policlinicogemelli.it (I.P.); luisa.carbognin@guest.policlinicogemelli.it (L.C.); giordana.tiberi@guest.policlinicogemelli.it (G.T.); francesco.pavese@guest.policlinicogemelli.it (F.P.); giovanni.scambia@policlinicogemelli.it (G.S.); 2Comprehensive Cancer Center, Unit of Medical Oncology, Fondazione Policlinico Universitario Agostino Gemelli IRCCS, 00168 Rome, Italy; dr.tatianadangelo@gmail.com (T.D.); armando.orlandi@policlinicogemelli.it (A.O.); giampaolo.tortora@policlinicogemelli.it (G.T.); 3Scientific Directorate, Fondazione Policlinico Universitario Agostino Gemelli IRCCS, 00168 Rome, Italy; simona.duranti@policlinicogemelli.it; 4Medical Oncology, Università Cattolica del Sacro Cuore, 00168 Rome, Italy; 5Istituto di Ginecologia e Ostetricia, Università Cattolica del Sacro Cuore, 00168 Rome, Italy; 6Precision Medicine in Breast Cancer Unit, Fondazione Policlinico Universitario Agostino Gemelli IRCCS, 00168 Rome, Italy; alessandra.fabi@policlinicogemelli.it

**Keywords:** PI3K inhibitors, subtype breast cancer, biomarkers, liquid biopsy, next-generation sequencing, real-time PCR, target therapy, clinical trial, precision medicine

## Abstract

**Simple Summary:**

Molecular aberrations in the phosphatidylinositol 3-kinase (PI3K) pathway are often observed in breast cancer and represent a key regulator of many cellular processes, promoting tumor cell growth and survival. The first clinical trials leading to the development of pan-PI3K inhibitors showed certain preclinical activity; nevertheless, the toxicity profile limited further analysis of this drugs’ class. To improve the antitumor effect and therapeutic index, additional clinical trials have been performed to develop the PI3K isoform-specific inhibitors and new schedule combinations with a good toxicity profile. However, further efforts are needed to discover other potentially actionable genetic alterations that remain a challenge to reach the goal of personalized and precision medicine.

**Abstract:**

Breast cancer is the leading cause of death in the female population and despite significant efforts made in diagnostic approaches and treatment strategies adopted for advanced breast cancer, the disease still remains incurable. Therefore, development of more effective systemic treatments constitutes a crucial need. Recently, several clinical trials were performed to find innovative predictive biomarkers and to improve the outcome of metastatic breast cancer through innovative therapeutic algorithms. In the pathogenesis of breast cancer, the phosphatidylinositol 3-kinase (PI3K)-protein kinase B (PKB/AKT)-mammalian target of rapamycin (mTOR) axis is a key regulator of cell proliferation, growth, survival, metabolism, and motility, making it an interest and therapeutic target. Nevertheless, the PI3K/AKT/mTOR cascade includes a complex network of biological events, needing more sophisticated approaches for their use in cancer treatment. In this review, we described the rationale for targeting the PI3K pathway, the development of PI3K inhibitors and the future treatment directions of different breast cancer subtypes in the metastatic setting.

## 1. Introduction

Breast cancer (BC) has been divided into four subgroups according to the different expressions of hormone and human epidermal growth factor (HER2) receptors: luminal A, luminal B, HER2 enriched, and basal-like. Each subtype is characterized by different clinical manifestations and prognosis [1,2,3,4]. The most advanced genomic assays have expanded BC molecular sequencing, shedding light on the detection of further genetic alterations and new potentially predictive and prognostic biomarkers [5].

Genetic dysregulation in the phosphoinositide 3-kinase (PI3K) pathway is frequently observed in BC. Alterations in the PI3K/AKT/mTOR pathway can either increase or abolish PI3K activity. Moreover, mutations in the tumor suppressing system, such as in the genes encoding for phosphatase and tensin homolog (PTEN) and inositol polyphosphate 4-phosphatase (INPP4B), can determine not only cancer growth, progression, survival, metabolism, protein synthesis and angiogenesis but also the risk of resistance to endocrine therapy and chemotherapy [6]. Therefore, a better understanding of the PI3K axis in BC in the metastatic settings is crucial, in order to better target the different molecular subtypes.

PI3Ks represents a group of heterodimeric lipid kinases at plasma and intracellular membranes characterized by catalytic (p110) and regulatory (p85) subunits. PI3K is divided into three classes with different biochemical properties and specificity. Class I PI3Ks, the most involved in BC, are divided by PI3Ks class IA, class IB, and class IC. The activation of the specific receptor—by the extracellular ligand—stimulates PI3K to catalyze phosphorylation of phosphatidylinositol 4,5-bisphosphate (PIP2) to phosphatidylinositol 3,4,5-triphosphate (PIP3), recruiting two central mediators of the PI3K cascade, the protein kinase B (AKT) and the phosphoinositide-dependent kinase 1 (PDK1). Then, the phosphorylation of AKT by PDK1 determines a series of different downstream signals mediated by the activation of mammalian target of rapamycin (mTOR), promoting the metabolic and proliferative setting of the cell. The tumor suppressors PTEN and INPP4B are negative regulators of AKT as they catalyze the dephosphorylation of PIP3 to PIP2, limiting the activation of the PI3K signaling axis [7,8,9] (Figure 1). 

PIK3CA is the main genomic aberration found in hormone receptor positive (HR+) BC, the mutation occurs in the encoding region of the p110α catalytic subunit of PI3K [4]. Other mutations involved in the PI3K cascade have a pathogenic role, even if less frequently found, as loss-of-function of PTEN (2–4%), activating mutations in AKT (2–3%) and in PI3K regulatory subunit α (1–2%). Similar alterations were found in HER2 positive (HER2+) BC. On the contrary, PTEN loss of expression and PIK3CA mutations can be found in triple negative breast cancer (TNBC) in about 30–50% and <10% of cases, respectively, conferring an unfavorable prognosis and resistance to standard of care (SOC) therapies [10].

The clinical implications of these mutations, however, still need to be clarified considering the controversial findings in the field of PIK3CA in BC [11,12,13]. Although a number of PI3K/AKT/mTOR axis inhibitors have been studied in pre-clinical settings, only a few reached the approval for the treatment of human cancer. In this review, we will provide an overview on the development and the most recent clinical findings of the different PI3K inhibitors in different breast cancer subtypes for the treatment of advanced BC.

## 2. Pan-PI3K Inhibitors 

Activating mutations in PIK3CA, the gene encoding for the p110α catalytic subunits of PI3K, are associated with the growth and survival of cancer cells that play a role in cell survival, proliferation, differentiation, and glucose transport [4,14,15,16]. Dysregulation of the PI3K-pathway may also contribute to resistance to a variety of anticancer agents [17,18]. First-generation PI3K inhibitors (PI3Ki), also known as pan-inhibitors of PI3K, target all four catalytic isoforms of class I PI3Ks (α, β, γ, and δ) [19,20]. These inhibitors encompass a broad spectrum of activities, and this broader inhibition leads to a higher risk of adverse events (AEs) and off-target toxicity, which have limited their use at therapeutic doses and caused treatment discontinuation [10,21,22].

### 2.1. Pan-PI3K Inhibitors in HR+/HER2− Breast Cancer Subtypes

In luminal BC subtype PIK3CA activating mutations occur in roughly 40% of cases and, in this setting, the pan-PI3Ki mostly include buparlisib (BKM120) and pictilisib (GDC-094) (Table 1).

#### 2.1.1. Buparlisib

Buparlisib (BKM120) is an orally selective pan-PI3Ki that inhibits all four class I PI3K isoforms (p110α, β, γ, and δ), as well as somatically mutated p110α, with no activity against the class III PI3K, or mammalian target of rapamycin (mTOR). In cellular assays and in vivo models of human cancers, buparlisib showed significant dose-dependent tumor growth inhibition, potent antiproliferative and proapoptotic activities [26,27]. Miller et al. reported that the hyperactivation of the PI3K pathway, or PTEN loss expression, promoted antioestrogen resistance in HR+/HER2− MBC, inducing ER-transcriptional independent activity and growth of BC cells [24,25]. Therefore, buparlisib plus fulvestrant has been successfully evaluated in a phase I study in postmenopausal women with HR+ MBC, previously treated with endocrine therapy [28]. PIK3CA mutations proved to be a biomarker of poor prognosis, such as PgR negativity, TP53 mutations, and loss of PTEN expression. Conversely, mutations in AKT1 and ESR1 did not prevent tumor response. Treatment-related adverse events (AEs) were generally mild and included hepatic toxicity, rash and hyperglycemia [29,30]. 

The phase III BELLE-2 clinical trial investigated the efficacy and safety of buparlisib plus fulvestrant in HR+/HER2− post-menopausal MBC patients, who were progressive during/after treatment with aromatase inhibitors (AI) and had received up to one prior line of chemotherapy [23]. Patients were randomized to receive fulvestrant plus buparlisib or placebo and stratified according to PI3K status in tumor tissue by Sanger sequencing, and visceral disease status. The trial met its primary endpoint demonstrating a relatively modest benefit in progression-free survival (PFS) (6.9 versus 5.0 months compared to placebo; HR 0.78 CI 0.67–0.89; *p* < 0.001). Many patients, in the buparlisib arm, discontinued the study early (1.9 months) due to adverse events (higher transaminase levels, hyperglycemia, rash, and mood disturbances) probably influencing the potential benefits of combination therapy. In the exploratory analysis on PIK3CA mutations identified in circulating tumor DNA (ctDNA), the combination treatment was associated with an improvement in PFS compared with fulvestrant alone only in patients with PIK3CA mutations. In contrast, prespecified analyses of PIK3CA mutation and/or loss of PTEN expression by Sanger sequencing in archival tissue samples did not show increased PFS with buparlisib, potentially suggesting a tumor evolution from initial diagnosis and treatment beginning.

Likewise, BELLE-3 trial explored the ability of fulvestrant plus buparlisib to restore endocrine sensitivity, compared to fulvestrant alone, in patients with luminal MBC subtype previously treated with ET and mTOR inhibitors [31]. The trial showed that the combination therapy was associated with better PFS (median 3.9 months versus 1.8 months; HR 0.67 CI 0.53–0.84; *p* < 0.001), particularly in patients with real-time polymerase chain reaction (RT-PCR) or ctDNA PIK3CA mutations. In ctDNA-detected PI3K alterations the median PFS was 4.2 months versus 1.6 months (HR 0.46 CI 0.29–0.73; *p* = 0.00031) while it was 4.7 months versus 1.4 months (HR 0.39 CI 0.23–0.65; *p* < 0.001) in the group tested by RT-PCR. On the contrary, in the BELLE-2 trial, patients with PIK3CA mutations tested in tumor tissue, PFS was not different between treatments. The discrepancy between the two studies could be explained by the different sensitivity of the test used for PIK3CA assessment, or by the absence of stratification, based on ctDNA at randomization and subsequently added as a secondary endpoint. 

Additionally, Baselga et al., in the combined analysis from BELLE-2 and BELLE-3 trials, observed that assessing PIK3CA-mutational status in ctDNA resulted in larger clinical benefits compared with tissue samples PCR [32,33]. The safety profile of buparlisib was consistent in BELLE-3 and BELLE-2. Psychiatric complications, related to the ability of buparlisib to penetrate the blood-brain barrier [34], such as depression, anxiety, and suicidal ideation, were reported in 2% of patients enrolled in the BELLE-3 trial, in contrast with the absence of suicidal ideation in BELLE-2 study. 

In phase II/III clinical trial BELLE-4 [35], the combination of paclitaxel plus buparlisib in MBC as first-line treatment was tested, showing that the addition of buparlisib to paclitaxel did not improve median PFS when compared to paclitaxel alone (8.0 months versus 9.2 months in buparlisib and placebo group, respectively; HR 1.18 CI 0.82–1.68). Furthermore, the HRs, in the group of patients with PI3K mutations, PTEN loss expression and PI3K wild type, were similar to the overall enrolled population. Buparlisib treatment was associated with a higher frequency of serious AEs (30.2% in the buparlisib arm versus 20.9% in the placebo group) and more than 40% of patients, in the combination arm, experienced diarrhea (55%), rash (43%), hyperglycemia, and nausea (41%).

#### 2.1.2. Pictilisib

Pictilisib (GDC-094) is an oral class I PI3K pan-inhibitor. In vitro experiments showed preliminary antitumor activity (equipotent inhibition of the p110α and -δPI3K isoforms and less potent inhibition of p110β and -γ isoforms) and, in human tumor xenograft murine models, pictilisib demonstrated a strong inhibitory effect on the growth of human U87MG glioblastoma and IGROV1 ovarian cancer [36].

In the dose-escalation phase I clinical trial, Sarker et al. [37] evaluated the preliminary clinical activity of pictilisib in sixty unselected and heavily pretreated patients with advanced solid tumors. A low response rate, consisting of one partial response (PR) and one stable disease (SD), was observed. The toxicity profile of pictilisib was similar to buparlisib and the most frequent grade 3/4 AEs were rash, hyperglycemia, and pneumonitis. In this study pictilisib-related hyperglycemia was limited to grade 1–2; grade > 3 hyperglycemia occurred only in one patient. Mood alterations were not significant due to the low central nervous system (CNS) penetration of pictilisib compared to Buparlisib. 

According to previous results by Schmid et al., in a phase II, open-label, randomized trial [38], the addiction of pictilisib to anastrozole determined a synergic effect and was associated with increased inhibition of tumor cell proliferation over anastrozole alone in postmenopausal women with luminal-B early BC. 

In the two-part, randomized, double-blind, placebo-controlled phase II clinical trial FERGI, [39] investigating the benefit of adding pictilisib to fulvestrant in postmenopausal patients with HR+/HER2− MBC resistant to AI, no difference in median PFS was found in the combination treatment arm, regardless of PI3KCA mutation status. The median PFS was 6.6 months and 5.1 months in the experimental arm and control arm, respectively (HR 0.74 CI 0.52–1.06; *p* = 0.096) in ITT-population and 6.5 months and 5.1 months, respectively (HR 0.73 95% CI 0.42–1.28; *p* = 0.268) in women with PIK3CA-mutations. Pictilisib treatment was affected by drug toxicity, potentially limiting its efficacy. Rash, diarrhea, increase transaminase level, hyperglycemia, pneumonitis, bronchopneumonia, pleural effusions, and fatigue were the most common AEs in the pictilisib arm and in approximately 60% of cases grade 3 or worse adverse events were observed, with serious AEs occurring in up to 16% of participants. 

Another phase II randomized placebo-controlled clinical trial, PEGGY [40], failed to meet the primary endpoint. This study, similar to BELLE 4, was conducted to evaluate the benefit of adding PI3K-inhibitors to paclitaxel vs. paclitaxel alone in pre-and post-menopausal HR-positive/HER2-negative advanced BC patients. Genetic alterations in the PIK3CA gene were assessed in about 35% and 32% of patients in both arms, respectively. At the interim analyses, no significant differences in terms of PFS and overall response (ORR) were observed, irrespective of PIK3CA mutation status. Indeed, the addition of pictilisib to paclitaxel did not improve median PFS in the entire (8.2 months vs. 7.8 months; HR 0.95 CI 0.62–1.46; *p* = 0.83) and in the PI3K pathway-activated study population (7.3 months vs. 5.8 months; HR 1.06 CI 0.52–2.12; *p* = 0.88). Toxicity due to pictilisib administration included maculopapular rash, hypertension, and hyperglycemia, and experimental treatment was also associated with higher grade ≥ 3 AEs, dose reduction, and discontinuation.

### 2.2. Pan-PI3K Inhibitors in HER2+ Breast Cancer Subtypes 

Although the anti-HER2 agents demonstrated strong efficacy in patients with HER2 + BC, de novo or acquired resistance to HER2-target therapies represents a wide field of research. About 25% of HER2 + BC harbor PIK3CA mutations confer resistance to anti-HER2 therapy and poorer prognosis [41,42,43,44,45,46]. Preclinical studies showed that HER2-signalling is extremely deregulated and is largely mediated by p110α rather than one of the other class-I PI3K isoforms, providing a robust rationale to target the PI3K pathway [47,48,49]. The tolerability and activity of buparlisib in combination with trastuzumab were investigated in a phase Ib/II dose-escalation trial in PIK3CA unselected HER2-positive BC, progressive to trastuzumab. The recommended dose of buparlisib was reached at 100 mg/day plus weekly intravenous trastuzumab but, even if evidence of clinical activity was observed, the study did not meet the pre-established primary endpoint such as objective response > 25% [50,51]. 

Furthermore, Guerin et al. studied buparlisib in combination with lapatinib, the orally dual anti-HER2/HER1 tyrosine kinase inhibitor, in the phase Ib/II trial PIKHER2, in HER2-positive trastuzumab-resistant MBC patients, independently to PIK3CA mutational status [52]. Twenty-four patients were treated at five different dose regimens; the selected dose was 80 mg/daily for buparlisib and 1000 mg/daily for lapatinib. Main drug-related adverse events leading to discontinuation of treatment were gastrointestinal disorders, skin rash, depression and anxiety. The disease control rate was 79% (95% CI 57–92%) including 4% of CR and a 29% of clinical benefit rate (CBR) (95% CI 12–51%).

### 2.3. Pan-PI3K Inhibitors in Triple Negative Breast Cancer Subtypes 

TNBC represents the most aggressive BC with an incidence of 15–20% and a high heterogeneity in the mutational profile. Due to limited possibility of target therapies, standard chemotherapy still represents the milestone of treatment [53]. Approximately 10% of TNBC harbors a germline mutation in BRCA1 or BRCA2 genes and recent evidence has shown that platinum-based therapy offers promising activity both in early and metastatic settings [54,55,56,57,58,59,60]. In TNBC, activating PIK3CA mutations, the majority located in the p110α subunit, are the second most frequent molecular aberrations after TP53 mutations and occur in 7–9% of primary TNBC, with a likely higher rate in advanced TNBC [1]. PIK3CA mutations, with additional inactivating alterations in PTEN [61] and activating mutations in AKT1, globally occur in about 25% of TNBC [62]. In this subtype, dysregulation of the PI3K pathway has been correlated with chemotherapy resistance [63] and loss of PTEN function confers resistance to PDL1-blockade and leads to increased PI3Kβ signaling. Indeed, based on preclinical models, combined therapy with an anti-PDL1 agent and a PI3Kβ inhibitor showed an improvement in cancer growth inhibition [62,64]. However, the role of the PI3K pathway-targeted therapy in TNBC is also unclear. The activation of the PI3K pathway is more linked with the androgen receptor-positive subtype of TNBC and less correlated to TNBC compared to HR-positive and HER2-positive BC [62,65]. In the BELLE-4 trial TNBC patients (about 25% of total) tended to have a worse prognosis with buparlisib versus placebo arm (5.5 versus 9.3 months; HR 1.86 CI 0.91–3.79) and versus HR+ population (9.2 vs. 9.2 months; HR 1.00 CI 0.66–1.529). In this trial, the poorer prognosis in the TNBC subgroup, may be explained by a lower duration of paclitaxel exposure in the buparlisib group suggesting that the toxicity of buparlisib may have compromised the adequate administration of chemotherapy; hence, the authors failed to confirm the PIK3CA mutation’s predictive role in TNBC subtype [35]. In preclinical models, the pan-PI3Ki BKM120 sensitized BRCA-proficient TNBC to PARP inhibitor olaparib: the authors demonstrated that the dual PI3K and PARP inhibition significantly downregulated BRCA1/2 expression and reduced tumor cell growth [66]. Likewise, a second study showed synergic activity of buparlisib combined with olaparib in an MMTV-CreBRCA1f/fp53 +/− mouse model of BC [67]. Additionally, Matulonis UA et al., in a phase I dose-escalation trial, demonstrated anticancer activity in BC (54% of which TNBC) and ovarian cancer and in both germline BRCA (gBRCA)-mutated and gBRCA-wild-type patients [68].

## 3. PI3K Isoform-Specific Inhibitors 

Even if buparlisib met the efficacy endpoint in patients with the PIK3CA mutation, particularly in combination with endocrine therapy, the toxicity profile observed in clinical trials limits its potential use such as to be adopted as SOC and determining the stop of further developments. Likewise, in the light of the similar poor tolerability of pictilisib (GDC-0941) in absence of significant difference in terms of outcomes, also in PIK3CA-mutant cohort, no additional clinical trials have been performed to develop this drug. On the other hand, the introduction of selective inhibitors of specific PI3K isoforms [2] allowed a safer profile and consequently a prolonged adherence at therapeutic doses, even if requiring a stricter selection of patients [10]. 

### 3.1. PI3K Isoform-Specific Inhibitors in HR+/HER2− Breast Cancer Subtypes 

In HR+/HER2− BC subtype, the most studied PI3K isoform specific inhibitors are alpelisib (BYL719) and taselisib (GDC-0032) (Table 2). 

#### 3.1.1. Alpelisib

Alpelisib (BYL719) is the first oral inhibitor to be approved by the US Food and Drug Administration (FDA) and by European Medicines Agency (EMA), specifically designed to target the p110α isoform of wildtype PI3Kα. The specificity determines a 50 times more potent action against PI3Kα than the other isoforms [48,73].

In preclinical models, Alpelisib inhibited the most common hotspot PIK3CA^H1047R^ and PIK3CA^E545K^ mutations at nanomolar concentration (4.6 nM/L) and interfered with PIK3CA-mediated downstream signaling; moreover, it induced a dose-dependent degradation of p110 protein levels in HR+/PIK3CA-mutated BC cell lines [33]. 

The phase I trial CBYL719X2101 (NCT01219699) included 134 heavily pre-treated patients with PIK3CA alterations across advanced cancer types and demonstrated sensitivity to the single agent alpelisib. At the achieved dose of alpelisib at 400 mg p.o. once daily or 150 mg p.o. twice daily, frequent treatment-related AEs of all grades were showed: hyperglycemia (51%), nausea (50%), skin rash (42.5%), decreased appetite (42%), diarrhea (40%), vomiting (31%), fatigue (30%), and mucositis (20%). A good activity was observed and the median PFS was 5.5 months with an improved benefit among PI3CA mutated BC [74]. 

A phase Ib study evaluated the MTD, the safety, and the efficacy of the combination of alpelisib plus fulvestrant in 87 heavily pre-treated luminal MBC and with about 60% of patients with PIK3CA alterations. The most frequent grade 3/4 AEs were hyperglycemia and maculopapular rash, and 10% of patients permanently discontinued. The indicated dose was 300 mg once daily. The ORr was 29% among patients with PIK3CA alterations compared to no objective tumor response in the PI3KCA wild-type group. The median PFS was longer (9.1 vs. 4.7 months) among patients with PI3KCA alterations than those with no PI3KCA mutations [75]. These preliminary findings motivated the following phase III SOLAR-1 randomized clinical trial (RTC) on the combination regimen with alpesib and fulvestrant.

The phase III randomized SOLAR-1 trial was conducted to assess the safety and the efficacy of alpelisib plus fulvestrant in contrast with only fulvestrant in man or post-menopausal women with luminal MBC who recurred or progressed on, or after AI treatment [21,69]. Pre-treatment with CDK4/6 inhibitors was permitted, but palliative chemotherapy or fulvestrant or mTOR inhibitors were not accepted as previous treatments. At a median follow-up of 30.8 months, in the PIK3CA-mutated group the median PFS was longer than control arm [11.0 months (95% CI 7.5–14.5) versus 5.7 months (HR 0.65; *p* < 0.001)]; in tumors with PIK3CA wild-type alpelisib did not show improvement in term of PFS (7.4 versus 5.6 months; HR 0.85 95% CI 0.58–1.25). Even if not statistically significant difference was shown in OS [39.3 months in the experimental arm (95% CI 34.1–44.9) versus 31.4 months (95% CI 26.8–41.3) in the fulvestrant arm (HR 0.86 95% CI 0.64–1.15 one-sided *p* = 0.15)], the absolute 8-month gain in the experimental arm was clinically relevant. Also, an increase of ORr was registered for patients receiving alpelisib/fulvestrant compared to control arm (26.6% versus 12.8%) showing mainly a PR [70]. Alpelisib demonstrated high incidence of hyperglycemia (64%), diarrhea (58%), nausea (45%), decreased appetite (36%), rash (36%) and maculopapular rash (14%) and higher permanent discontinuation due to AEs occurred in 25% of patients in the experimental group respect to fulvestrant group. 

As shown in the SOLAR-1 study, a high concordance of PI3KCA status between primary and metastatic site was documented. Through liquid biopsy by plasma ctDNA, authors found a median PFS for the alpelisib/fulvestrant in the ctDNA-defined PIK3CA-mutant cohort versus the control arm (10.9 months vs. 3.7 months, respectively) the presence of PIK3CA mutations identified in ctDNA resulted in a risk of progression of 45% versus 35% in patients with PIK3CA mutations evaluated in tissue samples [76,77]. Giving the easy access of ctDNA and the results of the trials showing a positive correlation of ctDNA-detected PIK3CA mutations [21,78], US FDA approved the liquid biopsy testing, recommending the possibility to verify the presence of PIK3CA mutations in the tumor tissue when ctDNA testing has resulted negative [33,69]. 

In an exploratory analysis, the tissue samples were retrospectively tested with FoundationOne CDx 324-gene panel by Next Generation Sequencing (NGS) and a total of 7% of patients presented tumor PIK3CA mutations identified by NGS but not by RT-PCR-based testing and, moreover, showed a benefit from alpelisib [79]; therefore, the FDA has authorized FoundationOne CDx as a companion diagnostic test.

B-YOND is an interesting phase 1b study that included HR+/HER2− premenopausal asian advanced BC patients treated with alpelisib or buparlisib with tamoxifen plus goserelin as first-line therapy. Despite the high percentage of discontinuation due to toxicity, alpelisib plus endocrine therapy may be a valuable resource for premenopausal women with HR+/HER2− MBC [80].

Recruitment for the SOLAR-1 started in 2015 when CDK4/6 inhibitors were not yet approved; therefore, only 6% of patients with PI3CA mutations, randomized in the experimental arm, had previously received cyclines [69]. 

In the post-cyclin-dependent kinase 4/6 inhibitor setting, the phase II, open label, BYLieve trial (NCT03056755) was designed to assess the efficacy and the safety of alpelisib plus endocrine therapy (fulvestrant or letrozole) in three cohorts defined by immediate previous treatment for patients with HR+, HER2− PIK3CA-mutations MBC. In this trial, 112 patients with centrally confirmed PIK3CA mutations in tumor tissue were enrolled based on their previous treatment (CDKi + AI, CDKi + fulvestrant, or systemic chemotherapy or endocrine therapy): enrollment was completed in prior CDKi + AI and CDKi + fulvestrant SOC cohorts. The cohort of patients with CDKi + AI as immediate prior therapy received oral alpelisib 300 mg/day (continuously) plus fulvestrant: 50% of patients showed no disease progression at 6 months with median PFS resulted in 7 months and ORr of 21% over a median follow-up of 12 months [72].

#### 3.1.2. Taselisib

Taselisib (GDC-0032) is an oral class I PI3K inhibitor; it is technically a beta-sparing inhibitor selectively inhibiting p110α, δ and γ isoforms with 30-fold lower potency against p110β [81]. Taselisib exhibits greater selectivity for mutant PI3Kα isoform, and it was expected to improve efficacy on PIK3CA-mutant tumors with better toxicity profile as compared with pan-PI3K inhibitors. In preclinical studies, treatment with taselisib described marked tumor suppressing effect on PIK3CA-mutant xenografts models [82]. 

A phase I dose-finding clinical trial demonstrated clinical activity of taselisib in patients with metastatic solid tumors, specifically in PIK3CA-mutant metastatic BC [83]. 

After these encouraging results, the SANDPIPER phase III trial showed the superiority of the combination of fulvestrant with taselisib over fulvestrant alone in postmenopausal HR+/PIK3CA-mutated MBC patients and progressive during/following AI [median PFS 7.4 months vs. 5.4 months (HR 0.7 95% CI 0.56–0.89; *p* = 0.004)], ORr 28% vs. 12%, *p* < 0.001). Conversely, among 120 participants with PI3KCA-wild type tumors, median PFS did not significantly differ between experimental arm versus control arm (5.6 versus 4.0 months; HR 0.69 95% CI, 0.44–1.08; *p =* 0.106). Further investigations of taselisib were stopped due to the high incidence of AEs, especially diarrhea (grade 3/4 of 12% versus <1% for hormonal therapy alone) and hyperglycemia (grade 3/4 of 11% for taselisib arm versus <1% for control arm) [71,84].

### 3.2. PI3K Isoform-Specific Inhibitors in HER2+ Breast Cancer Subtypes 

In HER2 + BC, the development resistance to SOC anti-HER2 antibody has been correlated with the constitutive activation of the PI3K pathway [85]. Moreover, in trastuzumab-resistant BC cell lines due to the PTEN loss expression, inhibition of PI3K pathway inhibits cell growth and overcomes trastuzumab resistance [86]. 

With the purpose of studying the overcoming resistant to trastuzumab by PI3K pathway, new associations were tested. Alpelisib combined to trastuzumab emtansine (TDM-1) in trastuzumab-resistant patients with loss of PTEN function, AKT overexpression or PIK3CA mutation, documented an ORR of 43% in all populations and 30% in TDM1-resistant patients. Although 59% of patients experienced grade 3 toxicity, AEs were globally manageable. These results suggest that some sort of activation of the downstream PI3K pathway might occur, potentially provoking a resistance effect to TDM-1, similar to trastuzumab [87,88].

MEN1611 is a novel orally available α-selective PI3K inhibitor. Preclinical data demonstrated activity either as single agent or in combination with targeted therapies as shown in both xenograft and patient-derived xenograft (PDX) models of BC bearing PI3Kα mutations. In HER2 + BC harboring PIK3CA-mutation, cell lines and PDX models, MEN1611 acted synergistically when combined with trastuzumab, induced depletion protein, and a pro-inflammatory phenotype [89,90].

### 3.3. PI3K Isoform-Specific Inhibitors in Triple Negative Breast Cancer 

The first in human phase I clinical trial studied the PI3KCB/PI3KCD inhibitor (AZD8186) in patients with PTEN-deficient/-mutated or PIK3CB-mutated/-amplified advanced TNBC in combination with abiraterone acetate or vistusertib (AZD2014), a dual mTOR inhibitor. AZD8186 showed significant anti-tumor activity in PTEN-deficient TNBC cell lines, especially when combined with anti-androgens or the dual mTORC1/2 inhibitor [91]. However, AZD8186 has limited efficacy as a single agent, but has improved efficacy when combined with paclitaxel and anti-PD1 in vivo [92]. 

Since upregulation of the PI3K/AKT/mTOR pathway has been related with resistance to microtubule-targeting drugs [93], alpelisib was combined with taxanes (paclitaxel or nab-paclitaxel) in two phase I/II trials. Both trials were interrupted for high grade toxicity in terms of hyperglycemia, grade 3 acute renal failure, and grade 4 leukopenia, in line with PEGGY and BELLE-4 studies [94,95,96].

Ashgar et al. showed that CDK4/6 and PI3 kinase inhibitor combinations have substantial activity in vitro and in vivo among PIK3CA mutant non-basal TNBC, both in the luminal androgen receptor and in mesenchymal-stem subgroups [97]. In luminal androgen receptor TNBC, high frequency PIK3CA mutations, provide rationale for the potential combination approach and reveal sensitivity to PI3K/mTOR inhibitors and AR targeted therapy [98]. A phase Ib clinical trial assessed the combination of taselisib with palbociclib in metastatic BC including a cohort of TNBC selected for activating PIK3 mutations and showed good treatment tolerability and promising preliminary anti-tumor activity [99]. 

Another preclinical study showed that blocking both PI3Kα and CDK4/6 has a synergic effect in multiple RB1-wild-type TNBC models, increasing apoptosis, cell-cycle arrest, and cancer immunogenicity. Particularly, the combination of PI3Kα and CDK4/6 inhibition, together with PD-1 and CTLA-4 inhibition, was able to induce the TNBC tumor regression in vivo providing a novel potential therapeutic approach for TNBC [100].

## 4. PIK3 Inhibitors: New Perspectives

Several clinical trials are ongoing to determinate the activity, efficacy, and toxicity profile of PIK3CA inhibitors in metastatic BC. These are mostly trials evaluating combination of PI3K inhibitors with different targeted agents to drive the treatment of different BC subtypes and to achieve the best possible disease control (Table 3).

### 4.1. PI3K Inhibitors in HR+ HER2− Breast Cancer Subtypes

Inavolisib (GDC-0077) is an oral experimental drug targeting and inhibiting the PI3K p110a catalytic subunit (NCY 03006172). Based on preclinical studies that proved synergy between CDK4/6 inhibitors and PI3K inhibitors, a phase I open label dose-escalation study tested GDC-0077 as a single agent or in combination with letrozole/fulvestrant and Palbociclib for the treatment of locally advanced or metastatic PIK3CA-mutant BC. Worthy of mention is the metformin-addition arm to prevent PIK3I-induced hyperglycemia. The trial showed that GDC-0077 plus CDK4/6I and endocrine therapy could be combined at maximum doses [104]. Hence, the combination therapy is being evaluated in the INAVO120 trial, a phase III, randomized, double blind, placebo-controlled study (NCT04191499) that enrolled endocrine resistant patients with PIK3CA-mutant/HR+/HER2−, locally advanced or MBC. The trial is designed to verify the progression free survival and safety of Inavolisib plus palbocilib and fulvestrant [101].

An ongoing multicentre, open-label, phase Ib study (NCT04355520) is evaluating the safety and efficacy of TQ-B3525, a novel selective oral PI3K α/δ inhibitor. The drug is combined with fulvestrant SOC in subjects with HR+/HER2− and PIK3CA mutation [102].

Moreover, a single arm open label pilot trial is ongoing to assess the combination of dapagliflozin, a sodium/glucose cotransporter-2 (SGLT-2)-inhibitor used to treat diabetes mellitus, with alpelisib + fulvestrant in patients with HR+/HER2− metastatic BC. The aim of this study is to determine if the addition of dapagliflozin to alpelisib and fulvestrant combination could lead to a significant reduction of all-grade hyperglycemia [103].

### 4.2. PI3K Inhibitors in HER2 + Breast Cancer Subtypes 

The combination of copanlisib, trastuzumab, and pertuzumab is currently tested in a phase Ib/II study for HER2+ stage IV BC with PIK3CA or PTEN mutation. Copanlisib is an intravenously administered PI3K inhibitor typically used to treat relapsed and refractory follicular lymphoma, but the addition of copanlisib to trastuzumab and pertuzumab in HER2 + BC therapy could increase the duration of response as compared to the standard treatment, overcoming the resistance due to the hyperactivation of PI3K signaling downstream of HER2. Incidence of AEs, dose limiting toxicities and PFS are the primary endpoints evaluated in this trial [105,117]. 

In the same setting of disease and in the same patient population a multicenter randomized, double-blind, placebo-controlled phase III study (NCT04208178) is evaluating alpelisib in combination with trastuzumab and pertuzumab as maintenance treatment after induction therapy with a taxane. The study is divided into multiple sections: part 1 is the open-label, safety run-in part of the study, designed to confirm the recommended phase III dose (RP3D) of alpelisib in combination with the monoclonal antibodies [106].

B-PRECISE-01 trial (NCT03767335) tests MEN 1611, a molecule targeting the p110 α and, to a lesser extent, β and γ isoforms. The primary outcome of this open-label, dose-escalation, phase Ib study is to verify the maximum tolerated dose while treatment emergent adverse events (TEAEs); PFS and OS are the secondary endpoints. MEN 1611 is investigated in association with trastuzumab in patients affected by Her2+ metastatic BC after progression from HER2 based therapy. In the same trial, fulvestrant is added in post-menopausal women with hormone-sensitive disease [107]. 

In the international, multicenter, open-label, controlled phase III RCT ALPHABET TRIAL (NCT05063786), the combination of alpelisib with trastuzumab +/− fulvestrant will be tested. Patients with HER2 + BC and documented PIK3CA mutation, previously treated with TDM-1, will be enrolled. The control arm with chemotherapy according to investigator RCT (vinorelbine, capecitabine or eribulin) makes this study particularly interesting [118]. 

### 4.3. PI3K Inhibitors in Triple Negative Breast Cancer Subtypes 

The purpose of EPICK-B3 (NCT04251533) is to determine whether treatment with alpelisib in combination with nab-paclitaxel is safe and effective in subjects with advanced TNBC. This RCT is divided into different parts: part A, that is a randomized, double-blind, placebo-controlled study for patients with PIK3CA mutation regardless of PTEN status; part B1, that is a single-arm, open-label study for patients with PTEN loss expression without PIK3CA mutation. If preliminary efficacy is demonstrated, study part B2 will explore alpelisib + nab-paclitaxel in the same patient population as part B1 in a randomized, double-blind trial [96,110].

Still in the TNBC, Mario-3 (Macrophage Reprogramming in Immuno-Oncology) trial (NCT03961698) is studying a “triple combination”, which includes the addition of a new investigational drug called eganelisib (IPI-549) to the FDA-approved combination treatment with nab-paclitaxel and atezolizumab. IPI-549 is an oral, selective inhibitor of PI3Kγ. that works by reprogramming M2 macrophages or myeloid derived suppressor cells (MDSCs) within the tumor microenvironment from a pro-tumor function to an anti-tumor function. These effects allowed decreasing immune-suppression and increasing immune-activation, leading to the activation and proliferation of T cells that can attack cancer cells [119,120]. The evaluation of CR according to RECIST v1.1 criteria is the primary endpoint of MARIO-3, while safety, ORr, time to CR (TTCR), time to response (TTR), duration of CR (DOCR), duration of response (DOR), and PFS are the secondary endpoints [111].

Two additional trials evaluate novel combinations of PI3Ki with immunotherapy: one trial analyzes the association of PI3Kγ inhibitor with nivolumab (NCT02637531), and another one evaluates the combination of PI3Kδ inhibitor with pembrolizumab (NCT02646748) [53,112,113].

Moreover, the phase I/II trial studies the side effects and best dose of copanlisib and its activity and efficacy when given together with eribulin in treating patients with TNBC (NCT04345913) [114].

Patients with luminal androgen receptor TNBC deserve a special mention. For this peculiar population a phase Ib/II trial (NCT02457910) is testing the side effects and best dose of taselisib when given together with enzalutamide, a potent androgen receptor (AR) currently used for the treatment of patients with metastatic castration-resistant prostate cancer [115].

Another phase I clinical trial is studying the association of the α-specific PI3-kinase inhibitor alpelisib with enzalutamide in PTEN loss function and AR+ TNBC (NCT03207529). However, enzalutamide, in a phase II study, has already demonstrated clinical activity and good tolerability in patients with advanced AR+ TNBC [21,116].

A promising combination strategy is represented by PARP/PI3K dual inhibitors. This combination synergically acts to block the growth of cancer cells: PI3K inhibition leads to downregulation of BRCA1/2 proteins which increase the degree of Homologous Recombination Repair (HRR) deficiency. Therefore, the purpose of an ongoing phase I trial (NCT04586335) is to assess the safety, tolerability, and preliminary efficacy of CYH33, a novel highly potent and selective oral inhibitor of PI3Kα, in combination with olaparib in BC and other solid tumors like ovarian, endometrial, or prostate cancer [108]. 

AZD8186 is a PI3Kβ inhibitor tested in a phase I trial (NCT03218826) in combination with docetaxel. This is essentially a dose escalation study in which patients with HR+/HER2− advanced BC or metastatic TNBC and PTEN or PIK3CB mutation are enrolled. The primary objectives are to determine MDT or RP2D and to assess the safety and tolerability of AZD8186 when administrated in combination with docetaxel. The possible mechanisms of acquired resistance to PI3Kβ inhibition will be also evaluated: sequencing data from pre- and post-treatment specimens of patients that initially responded to PI3Kβ inhibitor AZD8186 will be compared to identify newly acquired mutations or deoxyribonucleic acid copy number alterations [109]. 

The results from these several trials could give us a better perspective on how PI3K inhibition impacts on TNBC patients’ management.

## 5. Conclusions

As previously mentioned, PI3K/AKT/mTOR signaling pathway is highly involved in cell differentiation, proliferation, energetic and glucose metabolism, apoptosis, motility, and angiogenesis. The high frequency of genetic aberrations in this pathway can result in tumorigenesis and cancer progression, as well as intrinsic and acquired resistance to available anti-tumor treatments for BC. PI3K mutations can be detected by tumor tissue and/or ctDNA in all BC subtypes and provide the rationale for the development of inhibitors targeting PI3K axis. The first pan-PI3K inhibitors combined with ET showed efficacy in PIK3CA-mutated patients, and their lack of selectivity determined high toxicity profile and limitation of the use in clinical practice. The subsequent development of the PI3K isoform-specific inhibitors with better tolerability profile in part overcame these concerns and provided new treatment chances for luminal PIK3CA mutations MBC, progressed after ET. The p110α-specific alpelisib was recently included in the arsenal of the treatments of HR+ MBC patients. However, the use of this agent in clinical practice requires better management of side effects, and a more suitable selection of patient population. In order to optimize the efficacy of these drugs, further clinical trials are needed to identify novel predictive biomarkers, and to elucidates the possible mechanism of intrinsic and acquired resistance. Moreover, clinical studies on new combinations of the PI3K isoform-specific inhibitors with ET, chemotherapy, CDK 4/6 inhibitors, immune checkpoint inhibitors, or novel target agents are still ongoing and could enhance the clinical benefit due to the possible synergistic action of an innovative treatment tailored to the specific BC subtypes and driven by PIK3CA aberrations. Another challenge will be to identify future new associations that do not increase the intrinsic toxicity of the PI3k inhibitor and associate with those of the companion molecules. Actually, no evidence of new methods has been developed to overcome the additive toxicity induced by combination therapies; possible studies on polymorphisms could identify genomic markers of predictivity to specific drug-related toxicities. Finally, genetic profiling of BC patients could lead to a better case selection, particularly for patients with poorer prognosis.

## Figures and Tables

**Figure 1 cancers-14-02161-f001:**
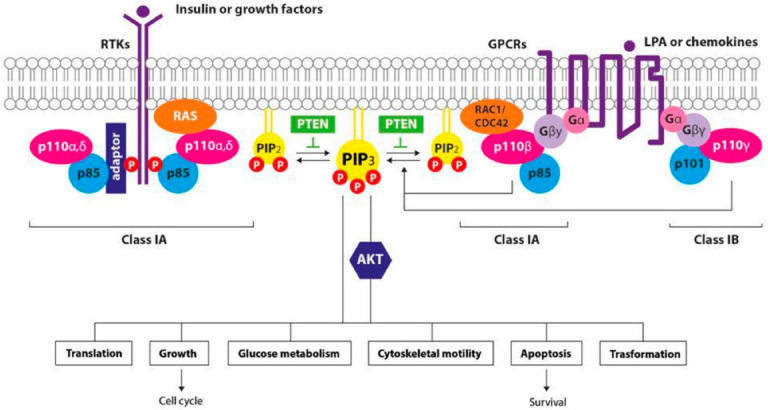
Signaling by PI3K isoforms in summary: growth factors bind their receptors on cell surface and activate the receptor tyrosine kinase (TK) or the G-protein coupled receptor (GPCR), promoting the recruitment of class I PI3Ks. The phosphorylation of PIP2 into PIP3 stimulates the downstream activation of both AKT-dependent and -independent signaling pathways. PTEN acts as regulator, removing the 3′ phosphate from PIP3 inactivating the signal perpetuated by class I PI3K [9].

**Table 1 cancers-14-02161-t001:** Summary of phases II-III trials with PI3K-inhibitors in HR-positive/HER2-negative metastatic breast cancer.

Trials	Phase	Patient Population	Targeted Therapy	Treatment	mPFS (Months)		mOS (Months)	ORr (%)
					ITT	Mutated PIK3CA	ITT	ITT
BELLE-2 (Baselga)[23] NCT01610284	III	HR+/HER2− LABC or MBC, resistant to AI	pan-PI3K inhibitor	Buparlisib + Fulvestrant	6.9 (HR 0.78)	7.0 (HR 0.58)	NA	11.8
				Placebo + Fulvestrant	5.0	3.2	NA	7.7
BELLE-3 (Di Leo)[18] NCT01633060	III	HR+/HER2− LABC or MBC, resistant to mTOR		Buparlisib + Fulvestrant	3.9 (HR 0.67)	4.2 (HR 0.46)	NA	8
				Placebo + Fulvestrant	1.8	1.6	NA	2
BELLE-4 (Martin)[21] NCT01572727	II-III	HER2− LABC or MBC		Buparlisib + Paclitaxel	8.0 (HR 1.18)	9.1 (HR 1.17)	NA	22.6
				Placebo + Paclitaxel	9.2	9.2	NA	27.1
FERGI (Krop)[24] NCT01437566	II	HR+/HER2− LABC or MBC, resistant to AI (only PIK3CA		Pictilisib + Fulvestrant	6.6 (HR 0.74)	6.5 (HR 0.73)	NA	7.9
				Placebo + Fulvestrant	5.1	5.1	NA	6.3
PEGGY (Vuylsteke) [25] NCT01740336	II	HR+/HER2− LABC or MBC first/second line CT		Pictilisib + Paclitaxel	8.2 (HR 0.83)	7.3 (HR 1.06)	NA	22
				Placebo + Paclitaxel	7.8	5.8	NA	19.6

Abbreviation: HR+: hormone receptor positive, HER2−: human epidermal growth factor receptor 2 negative, ABC: advanced breast cancer, LABC: locally advanced breast cancer; MBC: metastatic breast cancer, AI: aromatase inhibitors, ITT: intention-to-treat, mPFS: median progression free survival (months), mOS: median overall survival (months), ORr: overall response rate, NA: not available, CT: chemotherapy.

**Table 2 cancers-14-02161-t002:** Summary of phases II-III trials with PI3K isoform specific-inhibitors in HR-positive/HER2-negative metastatic breast cancer.

Trial	Phase	Patient Population	Targeted Therapy	Treatment	mPFS (Months)		mOS (Months)	Orr (%)
					Wild Type PI3KCA	Mutated PIK3CA	Mutated PIK3CA	Mutated PIK3CA
SOLAR-1 (Andrè) [69,70] NCT02437318	III	HR+/HER2− MBC, resistant to AI	PI3Kα inhibitor	Alpelisib + Fulvestrant	7.4 (HR 0.85)	11.0 (HR 0.65)	39.3 (HR 0.86)	26.6
				Placebo + Fulvestrant	5.6	5.7	31.4	12.8
SANDPIPER (Baselga)[71] NCT02340221	III	HR+/HER2− LABC or MBC, resistant to AI		Taselisib + Fulvestrant	5.6 (HR 0.69)	7.4 (HR 0.70)	NA	28
				Placebo + Fulvestrant	4.0	5.4	NA	11.9
BYLieve (Rugo)[72] NCT03056755	II	HR+/HER2− PIK3CA-mutated MBC, after CDKi + ET or CT or ET		Alpelisib + AI +/− LHRHa	-	NA	NA	NA
				Alpelisib + Fulvestrant +/− LHRHa	-	7.5	NA	21

Abbreviation: HR+: hormone receptor positive, HER2−: human epidermal growth factor receptor 2 negative, LABC: locally advanced breast cancer; MBC: metastatic breast cancer, AI: aromatase inhibitors, ITT: intention-to-treat, mPFS: progression free survival (months), mOS: overall survival (months), ORR: overall response rate, NA: not available, CDKi: CDK 4/6 inhibitor, ET: endocrine therapy, CT: chemotherapy, LHRHa: luteinizing hormone-releasing hormone agonist.

**Table 3 cancers-14-02161-t003:** Summary of ongoing phases I–III trials with PI3K-inhibitors in metastatic breast cancer according to different molecular subtypes.

BC Subtypes	Trials	Phase	Patient Population	Targeted Therapy	Treatment	Primary Endpoint	Secondary Endpoint
HR+/ HER2−	NCT04191499 [101] (INAVO 120)	II/III	400	PI3Kα-inhibitor	Inavolisib (GDC-0077) Palbociclib Fulvestrant	PFS	ORR BOR DOR CBR OS TTD AE
	NCT04355520 [102]	I/II	42	PI3K α/δ-inhibitor	TQ-B3525 Fulvestrant	DLT	ORR DCR DOR PFS OS
	NCT05025735 [103]	II	25	PI3Kα-inhibitor	Alpelisib Dapagliflozin Fulvestrant	Incidence of all grade hyperglycemia	Incidence of grade 3/4 hyperglycemia ORR PFS
HER2+	NCT03006172 [104]	I	256	PI3Kα-inhibitor	Inavolisib (GDC-0077) Fulvestrant Letrozole Palbociclib Metformin Trastuzumab Pertuzumab	DLTs	AUC Half-Life Cmax Cmin Time to Cmax of Inavolisib Apparent Clearance of Inavolisib ORR CBR DOR PFS
	NCT04108858 [105]	Ib/II	102	pan-PI3K—PI3K α/δ-inhibitor	Copanlisib Petruzumab Trastuzumab	AEs and SAEs DLTs PFS	PFS OS AEs and SAEs
	NCT04208178 [106]	III	588	PI3Kα-inhibitor	Alpelisib Pertuzumab Trastuzumab	DLTs PFS	OS ORR CBR TTR DOR
	NCT03767335 (B-PRECISE 01)[107]	I	48	PI3K α/β/γ-inhibitor	MEN1611 Trastuzumab Fulvestrant	MTD	TEAEs OS PFS
	NCT05063786 (ALPHABET)1 [9]	III	300	PI3Kα-inhibitor	Alpelisib Trastuzumab Fulvestrant Vinorelbine/ Capecitabine or Eribulin	PFS	OS OR Safety profile
HR+/ HER2 and TNBC	NCT04586335 [108]	Ib	350	PI3Kα-inhibitor	CYH33 Olaparib	DLT ORR	AEs DCR Pharmacokinetic measures
	NCT03218826 [109]	I	58	PI3Kβ-inhibitor	AZD8186 Docetaxel	MTD AEs	ORR CBR Drug—Drug interaction
TNBC	NCT04251533 (EPIK B3)[110]	III	566	PI3Kα-inhibitor	Alpelisib Nab-paclitaxel	PFS ORR	OS CBR ORr TTR DOR
	NCT03961698 (MARIO-3)[111]	II	90	PI3Kγ-inhibitor	IPI-549 (eganelisib) Nab-paclitaxel Atezolizumab	CR	TEAEs SAEs AEs ORr TTR DOR PFS
	NCT02637531 [112]	I	219	PI3Kγ-inhibitor	IPI-549 (eganelisib) Nivolumab	DLT AEs	AEs and safety laboratory Values Plasma concentrations of IPI-549 ORr CR PR DOR PFS OS
	NCT02646748 [113]	Ib	159	PI3Kδ-inhibitor	INCB050465 Pembrolizumab	Safety and tolerability profile	ORr Change in the number of TILs and the ratio of CD8+ lymphocytes to FOXP3+ cells infiltrating tumor post-treatment versus pretreatment by IHC
	NCT04345913 [114]	I/II	18	pan-PI3K-inhibitor / PI3K α/δ-inhibitor	Copanlisib Eribulina	MTD PFS	ORR CBR PFS
TNBC AR+	NCT02457910 [115]	I/II	30	PI3K α/δ/γ- inhibitor	Taselisib Enzalutamide	CBR MTD	PFS Pharmacokinetic profile
	NCT03207529 [116]	I	28	PI3Kα-inhibitor	Alpelisib Enzalutamide	MDT RP2D	DLT Safety profile

Abbreviation: AUC: Area Under the Concentration Time-Curve, Cmax: Maximum Plasma Concentration, Cmin: Minimum Plasma Concentration, AEs: adverse events, SAEs: serious adverse events, PFS: progression free survival, OS: overall survival, DLT: dose limiting toxicities, ORr: overall response rate, CBR: clinical benefit rate, TTR: time to response, DOR: duration of response, TEAEs: treatment-emergent adverse events, BOR: best overall response rate, TTD: time to deterioration, MTD: maximum tolerated dose, RP2D: recommended phase II dose, CR: complete response, TILs: Tumor Infiltrating Lymphocytes, IHC: immunohistochemistry.

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
