# Peer review of "PI3K Inhibitors in Advanced Breast Cancer: The Past, The Present, New Challenges and Future Perspectives"

_cancers, 2022, doi:10.3390/cancers14092161_

Round 1
Reviewer 1 Report
Reviewers Comments:
The authors present an interesting study in the review article entitled “PI3K inhibitors in Advanced Breast Cancer: The Past, The Present, New Challenges and Future Perspectives”. There are several review articles related with PI3K inhibitors in cancer. But the current review article really outstanding and covered most of the PI3K inhibitors developed against different subtypes of breast cancer. It would be worthwhile making some minor adjustments in the article for acceptance in “ Cancers Journal”.
Comment :
1) Author mentioned in the conclusion that “ New combinations of the PI3K isoform-specific inhibitors with ET, chemotherapy, CDK 4/6 inhibitors, immune checkpoint inhibitors, or novel target agents are still ongoing and could enhance the clinical benefit due to the possible synergistic action of an innovative treatment tailored to the specific BC subtypes and driven by PIK3CA aberrations”.
However, most of the PI3K inhibitors mentioned in this article demonstrated profound toxicity in the breast cancer patients. Still researchers are trying for combinational therapy to improve the therapeutic efficacy. Are their any methods developed to overcome the additive toxicity due to multiple combination treatments?
Author Response
Author mentioned in the conclusion that “New combinations of the PI3K isoform-specific inhibitors with ET, chemotherapy, CDK 4/6 inhibitors, immune checkpoint inhibitors, or novel target agents are still ongoing and could enhance the clinical benefit due to the possible synergistic action of an innovative treatment tailored to the specific BC subtypes and driven by PIK3CA aberrations”.
However, most of the PI3K inhibitors mentioned in this article demonstrated profound toxicity in the breast cancer patients. Still researchers are trying for combinational therapy to improve the therapeutic efficacy. Are there any methods developed to overcome the additive toxicity due to multiple combination treatments?
Response to reviewer 1 comments. We thank the referee for the comment. We agree and in the Paragraph 5 - “Conclusion” pag 17 this sentence was added: A challenge will also be to identify future new associations that do not increase the intrinsic toxicity of the PI3k inhibitor and associate with those of the companion molecules. Actually, no evidence of new methods has been developed to overcome the additive toxicity induced by combination therapies; possible studies on polymorphisms could identify genomic markers of predictivity to specific drug-related toxicities.
English language revised was done.

Reviewer 2 Report
The authors showcase clinical trials to observe PI3K in advanced BC. The study uses a combination of different drugs for different stages of BC. The study is comprehensive, well written, and may gain the interest of the scientific community. I highly endorse this study. but I think it is nice to include more recent studies for BC prognosis biomarkers in 2022. I suggest highlighting the following recent study (Pubmed:35205681).
Round 2
Reviewer 1 Report
The authors of the current review article have appropriately answered the comments from the reviewer. This article is ready for acceptance in the "Cancers" Journal.